# Effects of Cryogenic- and Cool-Assisted Burnishing on the Surface Integrity and Operating Behavior of Metal Components: A Review and Perspectives

**Jordan Maximov *** and **Galya Duncheva**

Department of Material Science and Mechanics of Materials, Technical University of Gabrovo, 5300 Gabrovo, Bulgaria; duncheva@tugab.bg
* Correspondence: jordanmaximov@gmail.com

**Abstract:** When placed under cryogenic temperatures (below −180 °C), metallic materials undergo structural changes that can improve their service life. This process, known as cryogenic treatment (CrT), has received extensive research attention over the past five decades. CrT can be applied as either an autonomous process (for steels and non-ferrous alloys, tool materials, and finished products) or as an assisting process for conventional metalworking. Cryogenic impacts and conventional machining or static surface cold working (SCW) can also be performed simultaneously in hybrid processes. The static SCW, known as burnishing, is a widely used environmentally friendly finishing process that achieves high-quality surfaces of metal components. The present review is dedicated to the portion of the hybrid processes in which burnishing under cryogenic conditions is carried out from the viewpoint of surface engineering, namely, finishing–surface integrity (SI)–operational behavior. Analyzes and summaries of the effects of cryogenic-assisted (CrA) burnishing on SI and the operational behavior of the investigated materials are made, and perspectives for future research are proposed.

**Keywords:** cryogenic treatment; cryogenic-assisted burnishing; surface integrity; roughness; microhardness; residual stresses; fatigue behaviour; wear resistance; corrosion resistance





## 1. Introduction

High-temperature treatments of metals have been performed since ancient times [1]. Such treatments are based on three fundamental properties of matter (adaptability, memory, and reflection) and consist of changing the temperature of metal components according to a corresponding law in order to obtain a desired structure. However, due to the aforementioned properties of matter, the structure of the metal components changes even at negative temperatures (in Celsius). The benefits of low-temperature effects on the structural modification of metallic components have only been realized and exploited during the last 100 years and are a direct consequence of advances in science and technology that enable the creation of ultra-low-temperature environments [2]. Previous studies [3] have shown that such approaches were already attempted in the first half of the last century, whereby negative temperatures were used to improve the operational behavior of watch parts, engine blocks, military aircraft components, and chainsaw blade links. Liquefied carbon dioxide was used as a coolant in machine operations as early as 1919 [2]. The term cryogenic treatment (CrT) was introduced in 1966, but systematic guidelines for the application of cryogenic engineering date from only 1976 [2]. Cryogenic is derived from the Greek word "κρύος" meaning cold. Cryogenic conditions have a very wide range of applications in modern engineering: to improve the properties and operational behavior of steels (martensitic [3–7] and austenitic [8–12]) and non-ferrous alloys [13,14], workpieces [15], finished components [3] and tools [16–18], and as a cooling medium for machining [19–22].

and plastic deformation [2]. It is, perhaps, for this reason of broad applicability that there is no consensus among researchers on temperature-based terminology.

Jawahir et al. [2], following the recommendations of research and standards organizations, indicated three limits (−150 °C, −153 °C, and −180 °C) below which cryogenic temperatures can be defined. The authors noted that −180 °C is a logical limit since the boiling points of the stable gases helium, nitrogen, neon, hydrogen, and oxygen are below −180 °C. Analyzing the processing of steels under negative temperature conditions, Diekman [4] applied the concepts of cold treating and cryogenic treatment, stating that the optimal temperature for cold treating is −84 °C, whereas typical CrT processes consist of a slow cool-down from ambient temperature to approximately −193 °C. In a comprehensive review paper dedicated to martensitic steels [3], the following concepts were summarized: cold treatment (>−80 °C); shallow CrT (from −80 °C to −160 °C); and deep CrT (<−160 °C); however, some authors using different cooling methods during machining have used the term cryogenic to denote any measured temperature from 0 °C to −100 °C [23–26]. It should be noted that the CrT is sometimes termed cryogenic processing, deep cryogenic processing, deep CrT, cryogenic tempering, and deep cryogenic tempering [4].

For cryogenic conditions (below −180 °C) [2], it is difficult to trace to which group of engineering objects (steels, tools, details, or processes) CrT was first applied. Figure 1 illustrates the application areas of CrT.

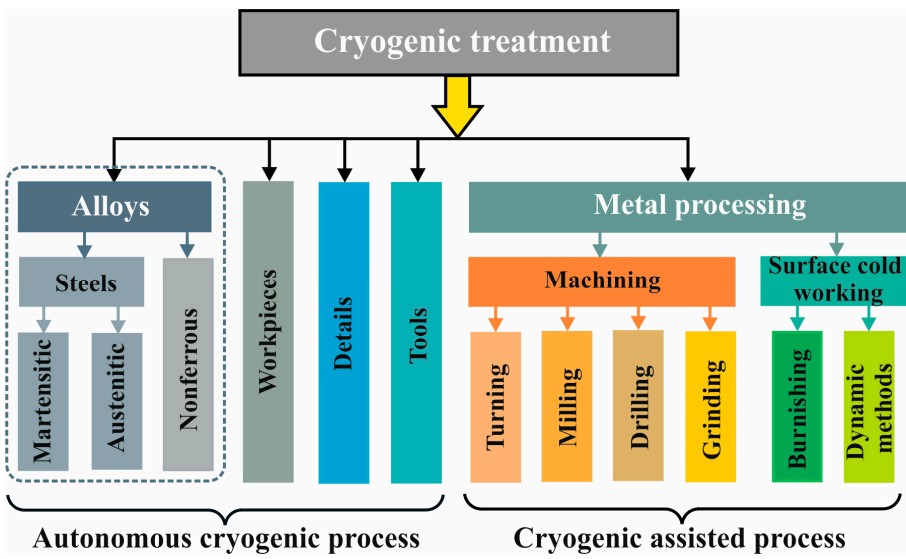

**Figure 1.** Fields of application of cryogenic treatment.

In summary, the benefits of applying CrT are as follows [2,4]: (1) relaxation of unwanted residual stresses; (2) strong reduction in residual austenite in martensitic steels; (3) precipitation of fine carbides in ferrous alloys; (4) improving the properties of steels and non-ferrous alloys by increasing their hardness and strength, dimensional stability, and thermal and electrical conductivity; (5) improvement of operational behavior in terms of increase in wear and corrosion resistances and fatigue life; (6) eliminations of damage induced by the heat, which is generated during machining and surface cold working processes.

As shown in Figure 1, CrT is applied as either an autonomous or an assisting process in conventional metalworking. Cryogenic-assisted (CrA) processes are based on promising directions of technology development involving different types of impacts. According to the way in which impacts are coordinated in time, two types of processes are possible: (1) hybrid processes, whereby the cryogenic impact and conventional machining are performed simultaneously; and (2) combined processes, in which impacts are applied sequentially. In practice, hybrid processes are implemented in two ways, according to the type of conventional machining used: (1) CrA machining (turning [2,25,27]; milling [2,23,28–30]; drilling [2,24]; grinding [2,26]); and (2) CrA burnishing [31–66]. Combined processes (CrA hard turn-

ing and subsequent burnishing) have been demonstrated in previous studies [15,67–70]. Chronologically, CrA machining precedes CrA burnishing. According to Jawahir et al. [2], the term cryogenic machining was first used by K. Uehara and S. Kumagai in 1968.

In summary, there are three types of cryogenic processes, illustrated in Figure 2: (1) autonomous CrT (see also Figure 1); (2) hybrid processes; and (3) combined processes.

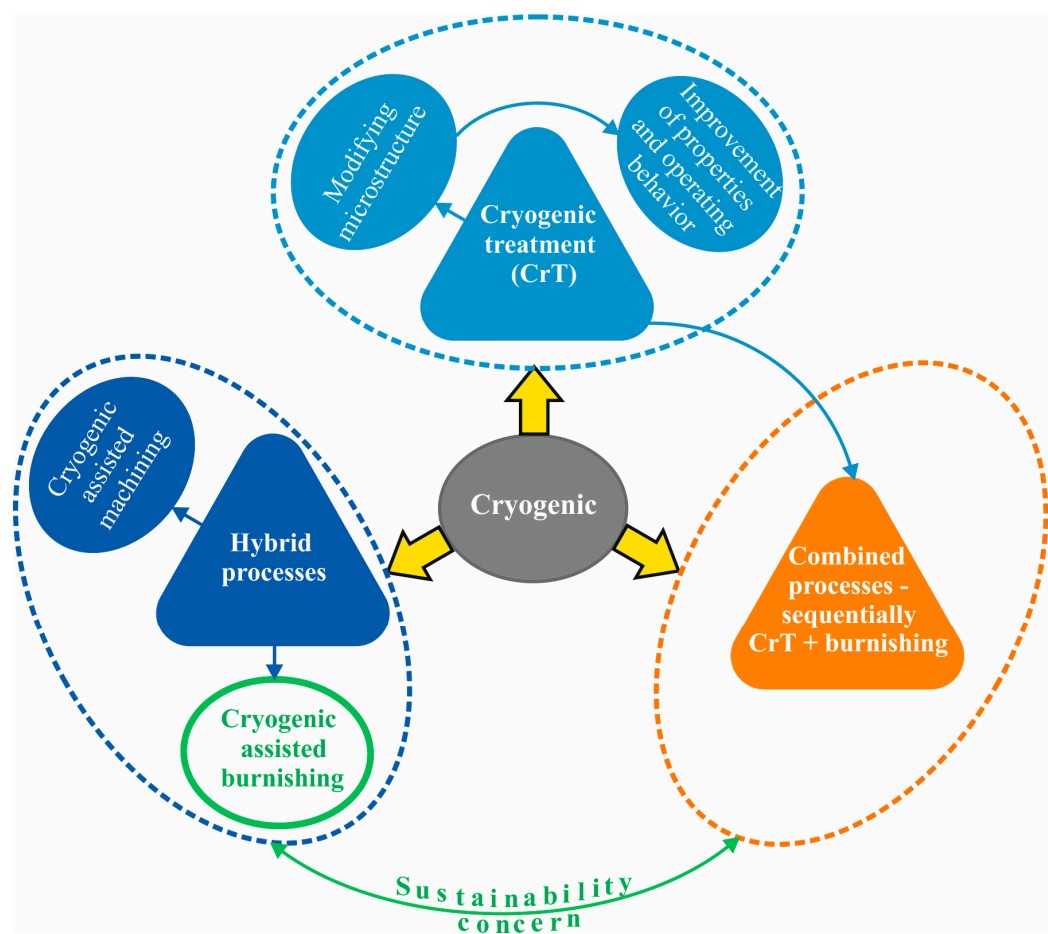

**Figure 2.** Types of cryogenic processes.

During the last two decades, numerous review papers have been published, primarily dedicated to (1) autonomous CrT of steels [3,71–73], tool materials [74], cutting tools [75], steels and non-ferrous alloys [76], and (2) CrA machining [2,77–80]. However, excluding Jawahir et al. [2], in which some attention is paid to burnishing, there is no comprehensive review paper dedicated to CrA burnishing.

Burnishing of metal components is a static surface coldworking method, which creates three effects [81]: smoothing, cold work, and residual compressive stresses at the surface and in adjacent subsurface layers. Following [82], the present work distinguishes between the concepts of the burnishing method and the burnishing process. A burnishing method is a coherent time–space mechanical arrangement of two bodies (the deforming element and the surface being treated) with a defined geometry and known physical and chemical properties. In contrast, the burnishing process is an energy–force exchange resulting from coherent interactions between the two bodies with clearly defined quantitative characteristics. Therefore, using the same method but with different magnitudes of the governing factors, many deforming processes can be undertaken, and, as a result, the processed components will have different surface integrity (SI) values.

Classifications of conventional burnishing methods have been made according to different characteristics [82]: (1) according to the shape of the deforming element (roller or ball), termed roller/ball burnishing (RB/BB); and (2) according to the type of contact

(sliding or rolling) between the deforming element and the treated surface, termed slide burnishing (SB) and RB/BB. Depending on the desired SI characteristics of the processed component, Ecorol [81] defined two processes: roller burnishing (RB); and deep rolling (DR). The primary goal of the first process is to produce mirror-like surfaces, while the accompanying beneficial effects of cold work (e.g., increased surface microhardness and compressive residual stress) are less pronounced. Conversely, the second process introduces significant compressive residual stresses and substantially increases the microhardness. Both processes are implemented either by the single-roller RB method or the BB method (including hydrostatic sphere). According to previous work [83,84], SB and diamond burnishing (DB) methods can implement three processes: smoothing; hardening; and mixed burnishing. Using the ball burnishing with hydrostatic sphere (BBHS) method, Lambda Research invented the low-plasticity burnishing (LPB) process [85], which was intended to create large compressive residual stresses at significant depths, whereby the equivalent plastic strain (degree of cold work) is automatically controlled in order to not exceed the set limit. All conventional processes can be converted into hybrid processes by means of cryogenic assistance. Figure 3 shows the schematics of CrA burnishing methods.

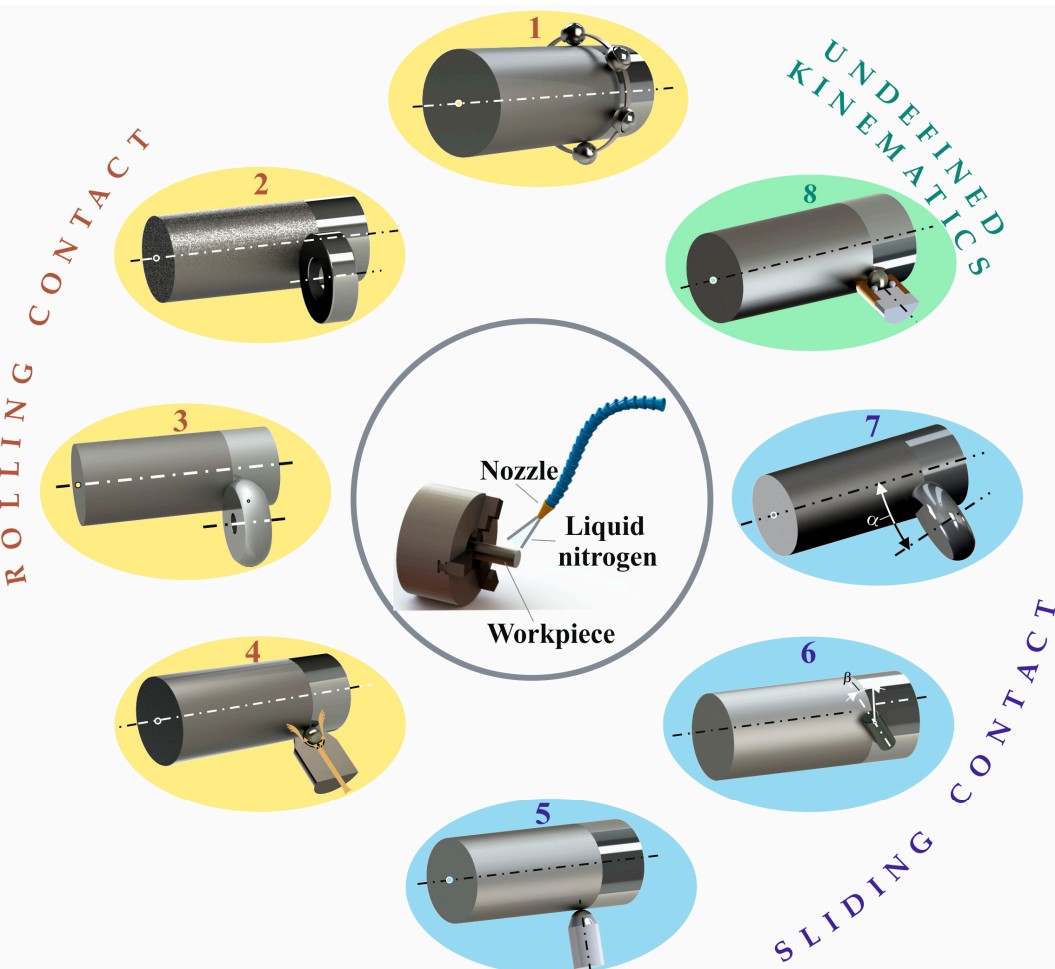

**Figure 3.** Schemes of cryogenic-assisted burnishing methods: 1—multiple ball burnishing; 2—roller burnishing with cylindrical roller; 3—roller burnishing with toroidal roller; 4—ball burnishing with hydrostatic sphere; 5—slide burnishing with spherical-ended insert; 6—slide burnishing with cylindrical-ended insert; 7—slide-roller burnishing; 8—ball burnishing with undefined ball motion.

This review paper is dedicated to the portion of the hybrid processes in which static surface cold working under cryogenic or cool conditions occurs, i.e., CrA or cool-assisted (CoA) burnishing. This review considers the viewpoint of surface engineering [86],

i.e., finishing–SI–operational behavior, with the aim of summarizing the effects of CrA and CoA burnishing on the SI characteristics and operational behavior of the investigated materials and proposing future research directions.

## 2. Investigations of CrA and CoA Burnishing—General Overview

In the present review, we follow the previous [2] in adopting the term cryogenic-assisted burnishing (CrAB) when the temperature in the surface plastic deformation zone is below −180 °C. At higher temperatures (for example, when using dry ice), the term cool-assisted burnishing (CoAB) is used. Although cryogenic machining [2] has been used since 1968, research on CrAB and CoAB has been performed more recently (2011–2024) and is significantly lower in volume than that dedicated to cryogenic machining. Much research on CrAB has been performed at the University of Kentucky, USA [32–36,42–45,60–66], although studies dedicated to CrAB and CoAB by other authors have also been performed worldwide [31,37–41,46–59]. Studies concerning CrAB and CoAB are systematized and classified in Tables 1 and 2.

**Table 1.** Investigations on hybrid processes—general overview.

| | Material | Burnish. Method/ Process | Agent | Hybrid Process | Other Cooling/Heating/Lubricating Conditions | | | | | Reference |
|---|---|---|---|---|---|---|---|---|---|---|
| | | | | | Flood | Dry | MQL | Hybrid (Cooling+ MQL) | Heating | |
| Magnesium alloys | AZ31B-O | SB | LN2 | CrA | | | | | | [42,43] |
| | AZ31B-O | SB | LN2 | CrA | | √ | | | | [44,45] |
| Titanium alloys | Ti-6Al-4V | RB/DR | LN2 | CrA | √ | √ | | | | [32] |
| | Ti-6Al-4V | RB/DR | LN2 | CrA | √ | √ | √ | √ | | [33] |
| | Ti-6Al-4V | RB/DR | LN2 | CrA | √ | | √ | √ | | [34] |
| | Ti-6Al-4V | RB/DR | LN2 | CrA | | √ | √ | | | [48,49] |
| | Ti-6Al-4V | SB | LN2 | CrA | | √ | | | | [57] |
| | Ti-6Al-4V | SB | LN2 | CrA | | | | | | [58] |
| Aluminum alloy | 7050-T7451 | RB/DR | LN2 | CrA | | √ | | | | [35,36] |
| Additively manufactured materials | Inconel 718/ Laser Powder Bed Fusion | RB/RB | LN2 | CrA | | √ | | | √ | [37] |
| | laser-clad Stellite 6 | BBHS/LPB | LN2 | CoA | | | | | | [59] |
| Carbon steel | SS 400 | SB/ Friction stir | CO2 | CoA | | √ | √ | | | [38,47] |
| | SS 400 | SB/ Friction stir | CO2 | CoA | | √ | √ | √ | | [46] |
| Tool steel | AISI D3 | BBHS/DR | CO2 | CoA | √ | | | | | [39] |
| | AISI D3 | BBHS/DR | CO2 | CoA | √ | | | | | [40] |
| Martensitic stainless steel | 17-4 PH | DB | LN2 | CrA | | √ | √ | | | [50,52] |
| | 17-4 PH | DB | LN2 | CrA | | | | | | [51,53–55] |

**Table 1.** *Cont.*

| | Material | Burnish. Method/ Process | Agent | Hybrid Process | Other Cooling/Heating/Lubricating Conditions | | | | | Reference |
|---|---|---|---|---|---|---|---|---|---|---|
| | | | | | Flood | Dry | MQL | Hybrid (Cooling+ MQL) | Heating | |
| Austenitic stainless steel | AISI 304 | BBHS/DR | LN2 | CrA | √ | | | | √ | [41] |
| Biomaterials | Cast homogenized Mg-4Zn-2Sr | BB/RB | LN2 | CrA | | | | | | [31] |
| | Co-Cr-Mo AZ31 | SB | LN2 | CrA | √ | | | | | [60] |
| | Co-Cr-Mo | SB | LN2 | CrA | √ | | | | | [61–63,66] |
| | Co-Cr-Mo | SB | LN2 | CrA | √ | | | | | [64] |
| | Co-Cr-Mo | SB | LN2 | CrA | | | | | | [65] |
| Thermal spray coating | | SB | LN2 | CrA | √ | | | | | [56] |

Abbreviation: BB—ball burnishing; BBHS—ball burnishing with hydrostatic sphere; DR—deep-rolling process; RB—roller-burnishing method (roller burnishing process); LPB—low-plasticity burnishing; LN2—liquid nitrogen; SB—slide burnishing; DB—diamond burnishing; CrA—cryogenic-assisted; CoA—cool-assisted. √ is an indicator of the presence of studies in the literature.

**Table 2.** Effect of hybrid processes on SI and operating behavior of materials.

| | Material | Burnish. Method/ Process | Hybrid Process | SI Characteristics | | | | Operating Behavior | | | Reference |
|---|---|---|---|---|---|---|---|---|---|---|---|
| | | | | R | MH | RS | M | F | WR | CR | |
| Magnesium alloys | AZ31B-O | SB | CrA | √ | √* | | √* | | | √* | [42,43] |
| | AZ31B-O | SB | CrA | | √* | | √* | | | √* | [44,45] |
| Titanium alloys | Ti-6Al-4V | RB/DR | CrA | √ | √* | | | | | | [32] |
| | Ti-6Al-4V | RB/DR | CrA | | √* | | √* | | | | [33] |
| | Ti-6Al-4V | RB/DR | CrA | √ | √* | | | | | | [34] |
| | Ti-6Al-4V | RB/DR | CrA | √ | √* | | | | √* | | [48,49] |
| | Ti-6Al-4V | SB | CrA | √* | | | √* | | | √* | [57] |
| | Ti-6Al-4V | SB | CrA | | | | √* | | | √* | [58] |
| Aluminum alloy | 7050-T7451 | RB/DR | CrA | | √* | | √* | | | | [35,36] |
| Additively manufactured materials | Inconel 718/ Laser Powder Bed Fusion | RB/DR | CrA | √ | √* | | | | | | [37] |
| | laser-clad Stellite 6 | BBHS/LPB | CoA | √* | √* | √* | | | | | [59] |

**Table 2.** *Cont.*

| Material | | Burnish. Method/Process | Hybrid Process | SI Characteristics | | | | Operating Behavior | | | Reference |
|---|---|---|---|---|---|---|---|---|---|---|---|
| | | | | R | MH | RS | M | F | WR | CR | |
| Carbon steel | SS 400 | SB/Friction stir | CoA | √* | | | | | | | [38,46] |
| | SS 400 | SB/Friction stir | CoA | | √* | | | | | | [47] |
| Tool steel | AISI D3 | BBHS/DR | CoA | √ | √* | | √* | | | | [39] |
| | AISI D3 | BBHS/DR | CoA | √ | √* | | √* | | | | [40] |
| Martensitic stainless steel | 17-4 PH | DB | CrA | | √* | √* | | | | | [50] |
| | 17-4 PH | DB | CrA | √* | √* | √* | √* | | | | [52] |
| | 17-4 PH | DB | CrA | √* | √* | | | | | | [51,53–55] |
| Austenitic stainless steel | AISI 304 | BBHS/DR | CrA | | | | √ | √ | √ | | [41] |
| Biomaterials | Cast homogenized Mg-4Zn-2Sr | BB/RB | CrA | √* | √* | | | | | | [31] |
| | Co-Cr-Mo AZ31 | SB | CrA | | √* | | √* | | | | [60] |
| | Co-Cr-Mo | SB | CrA | | √* | | √* | | | | [61,63] |
| | Co-Cr-Mo | SB | CrA | √* | √* | | √* | | | | [62,66] |
| | Co-Cr-Mo | SB | CrA | | √* | | | | √* | | [64] |
| | Co-Cr-Mo | SB | CrA | | | | √* | | | | [65] |
| Thermal spray coating | | SB | CrA | √* | √* | √* | √* | | | | [56] |

Abbreviation: BB—ball burnishing; BBHS—ball burnishing with hydrostatic sphere; DR—deep-rolling process; RB—roller-burnishing method (roller burnishing process); LPB—low-plasticity burnishing; SB—slide burnishing; DB—diamond burnishing; CrA—cryogenic-assisted; CoA—cool-assisted; R—roughness; MH—microhardness; RS—residual stresses; M—microstructure; F—fatigue behavior; WR—wear resistance; CR—corrosion resistance. √ is an indicator of the presence of studies in the literature. √* is an indicator of improvement after CrA (CoA) hybrid processes.

Figure 4 illustrates the percentage share of the investigated materials. CrAB and CoAB are most commonly applied to four types of steel, followed by three types of biomaterials. The effect of hybrid processes on coatings has received the least investigation. It is important to note that there is a general lack (with one exception [35,36]) of research on the application of high-strength aluminum alloys in the aerospace industry. The SB method (Figure 5), using either a diamond or non-diamond deforming element, is most often used, followed by the RB and BBHS methods. The latter methods are used to implement DR processes. In 86.11% of cases, the hybrid CrA process is implemented using liquid nitrogen. In the remaining cases, the hybrid CoA process is carried out using dry ice as the cooling medium. Figure 6a shows the percentage share of SI characteristics investigated in previous research. The most frequently studied feature is microhardness, followed by microstructure; in contrast, the least attention has been paid to the effects of the hybrid process on residual stresses. Limited research has been devoted to the influence of CrA and CoA processes on operational behavior (Figure 6b), whereas fatigue was only investigated in [41].

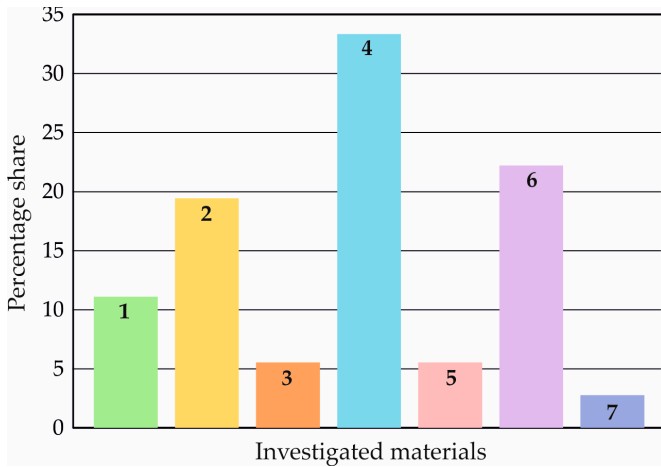

**Figure 4.** Percentage share of the studied materials: 1—magnesium alloys; 2—titanium alloys; 3—aluminum alloys; 4—steels; 5—additively manufactured materials; 6—biomaterials; 7—thermal spray coatings.

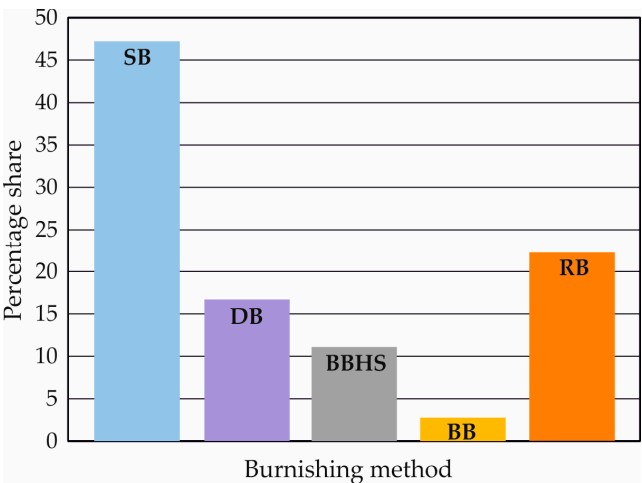

**Figure 5.** Percentage share of the used burnishing methods: SB—slide burnishing; DB—diamond burnishing; BBHS—ball burnishing with hydrostatic sphere; BB—ball burnishing; RB—roller burnishing.

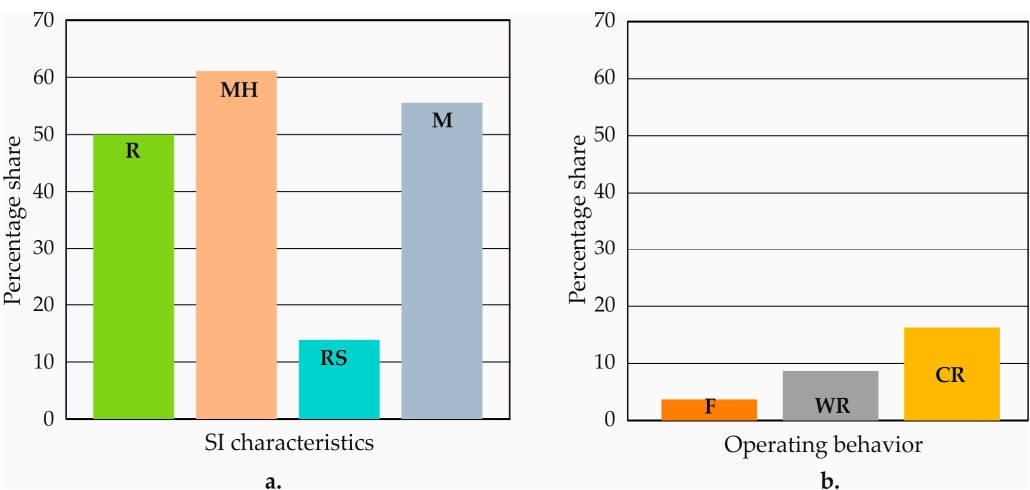

**Figure 6.** Percentage share of SI characteristics (**a**) and operating behavior (**b**): R—roughness; MH—microhardness; RS—residual stresses; M—microstructure; F—fatigue; WR—wear resistance; CR—corrosion resistance.

## 3. Effects of CrA and CoA Burnishing on SI

The effects of hybrid processes on SI characteristics are analyzed for each of the studied groups of materials.

### 3.1. Magnesium Alloy AZ31B-O

The effect of CrA SB implemented by high-speed steel cylindrical-shaped deforming elements on the SI characteristics of cylindrical specimens was studied by Pu et al. [42–45], who established that grinding at the room temperature provides a lower value of the Ra roughness parameter (Ra = 0.18 μm) than CrA SB (Ra = 0.22 μm). Due to CrA SB, the surface microhardness (as well as the microhardness in the layers adjacent to the surface) increases to 1.35 GPa, and the bulk material microhardness reaches 0.9 GPa. In terms of residual stresses, Pu et al. [42–45] established the following: (1) CrA SB introduces positive hoop stresses; and (2) the axial stresses introduced are compressive but are smaller in absolute value than those introduced via dry SB. The explanation for the positive circumferential stresses lies in the relatively large tangential force applied (500 N). This force is known to be minimal in the RB process. The CrA SB homogenizes and strongly grinds grains in the surface and near-subsurface layers, and cryogenic temperatures suppress grain growth due to dynamic recrystallization. Grain refinement is carried out by a mechanism of dynamic recrystallization.

The SI characteristics of the same magnesium alloy were also investigated by Yang et al. [60], who implemented CrA SB using a carbide fixed roller as the deforming element and employed an outer cylindrical treated surface. Comparing with the results achieved by conventional dry SB, the authors have found that CrA SB provides (1) significantly higher microhardness up to a depth of approximately 230 μm and (2) remarkably refined microstructures on the burnished surface (grains smaller than 300 nm were observed).

### 3.2. Titanium Alloy Ti-6Al-4V

The effects of CrA hybrid processes on the SI of this alloy were studied by Caudill et al. [32–34], Rotella et al. [48,49], and Tang et al. [57,58].

#### 3.2.1. Roughness

Starting from an initial roughness of Ra = 1.31 μm (after turning), Caudill et al. [32] found that the CrA DR process achieved a smaller reduction (56.7%) of initial roughness than that achieved by dry (63.4%) and flood-cooled (58.2%) processes. All three processes are realized by means of the RB method using a 3.04 mm diameter tungsten carbide roller and four magnitudes of burnishing force. The influence of the burnishing force on roughness was found to be less than that of the burnishing conditions. Figure 7 summarizes the effects of CrA DR on the investigated characteristics of SI [32]. To compare the change trends of heterogeneous quantities, the Ra roughness parameter and the values of microhardness are normalized to unity, i.e., the maximum value of the corresponding quantity is taken as unity.

In a previous study [34], it was shown that CrA and flood-cooled DR processes, implemented by a burnishing force of 1500 N, achieved a reduction in roughness by 61% and 64.8% of the initial value, respectively. The increased surface tension and greater friction explain the higher values of the roughness parameter Ra obtained by the CrA DR process compared to conventional flood-cooled DR. It was found that when the number of tool passes is greater than seven, the trend reverses. Rotella et al. [48,49] reported higher values of the integral height parameter Ra achieved by the CrA DR process compared to DR under the minimum quantity of lubricant (MQL) condition; however, the values observed were lower than those with the dry DR process. Tang et al. [57,58] implemented the CrA SB process using a cemented carbide deforming cylindrical-shaped element, whose axis was orthogonally crossed with the axis of the rotating cylindrical workpiece. The resulting roughness was Ra = 0.32 μm. Dry SB was found to significantly degrade the roughness.

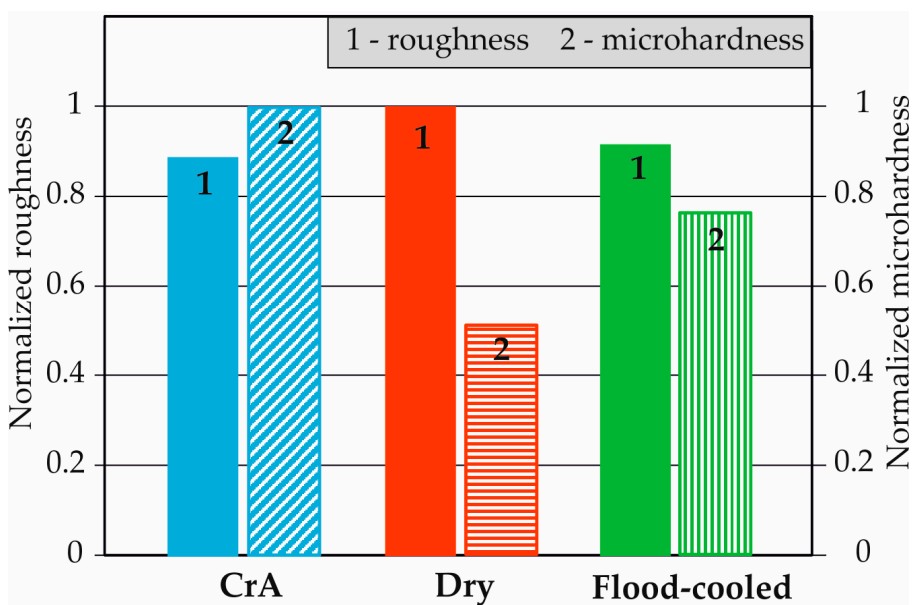

**Figure 7.** Comparison of the effects of DR on the roughness and microhardness of Ti-6Al-4V under different cooling conditions.

### 3.2.2. Surface Microhardness

Caudill et al. [32] identified a significant increase in surface microhardness due to CrA DR treatment relative to DR under dry and flood-cooled conditions for all investigated magnitudes of burnishing force (1000, 1500, 2000, and 2500 N). The maximum increase relative to turning in each case was as follows: CrA DR—64.2%; flood-cooled DR—48.9%; and dry DR—32.8% (see Figure 7).

CrA DR resulted in the greatest depth of the hardened layer, followed by DR under flood-cooled and dry conditions, respectively. The explanation for this phenomenon is as follows. The presence of a cooling mechanism in lubricated DR and CrA DR compensates for the thermal softening effect observed in dry DR. The coolant removes the heat generated during plastic deformation, allowing for the material to be compressed at higher burnishing forces. As the force increases, the density of dislocations in the surface and subsurface layers increases, leading to strain hardening. Therefore, two opposing mechanisms, namely, strain hardening and thermal softening, act on the material. Without coolant, thermal softening predominates over strain hardening because the heat accumulates near the surface, essentially annealing the dislocation mesh and reducing the hardness. Conventional flood cooling removes some of the generated heat, but liquid nitrogen provides a significantly more efficient coolant. Therefore, CrA surface plastic deformation is the most effective method to increase surface microhardness due to the strong cooling effect of liquid nitrogen.

A previous study [33] established that implementing the CrA DR process (using a pneumatically actuated tool equipped with a tungsten carbide roller with a 1.5 mm radius of curvature) in combination with MQL, provides maximum surface microhardness, i.e., higher than any of the following burnishing conditions: dry; MQL; flood-cooled; and cryogenic. It was found that increasing the number of passes (up to eight) increases the microhardness under all burnishing conditions. For example, eight-pass CrA DR with 1500 N burnishing force increases the microhardness by 45% relative to the initial state. Almost the same increase is obtained from single-pass CrA DR with a 2500 N burnishing force. On this basis, multipass CrA DR implemented with less force has been found to be a suitable finishing for parts with lower bending stiffness, such as turbine and compressor blades [33]. A comparison has also been made between the effects of the DR process, implemented with a burnishing force of 1500 and 2500 N, on the surface microhardness for four types of burnishing conditions: MQL; flood-cooled; cryogenic; and cryogenic + MQL, depending on the feed rate [34]. For all combinations of burnishing

parameters, the cryogenic and cryogenic + MQL conditions provided the highest surface microhardness and the greatest depth of the affected layer. The increase in surface microhardness is approximately 41% more than the initial state. A similar result was reported by Rottela et al. [48,49] by implementing the DR process under three burnishing conditions (dry, MQL, and cryogenic), showing that the optimal surface microhardness and depth of the affected layer were provided by the CrA DR process.

### 3.2.3. Microstructure

Caudill et al. [33] established that the CrA DR process creates a nanostructured layer with a thickness of 1 μm and grain sizes ranging from 51 to 212 nm, causing the surface microhardness to increase. Below this layer, the grain refinement becomes increasingly less pronounced with depth until the microstructure transitions to its bulk state. Tang et al. [57] reported that the grain size in the nanostructured layer obtained by single-pass CrA SB is 24.3 nm, and the length ratio is 1.74. Using a two-pass process, the grain size can be reduced to 20.7 nm while the length ratio is reduced to 1.45.

### 3.3. Aluminium Alloy 7050-T7451

The effects of the CrA DR process on the SI characteristics of this alloy have been investigated by Huang et al. [35,36], who found that the CrA process increased surface microhardness by an average of 20–30% within a layer of 200 μm depth; this is more significant than the 5–10% increase when using dry conventional burnishing. The authors' explanation is that tangential burnishing forces occurring during the CrA process are higher than those of the dry conventional burnishing due to rapid cooling and material work hardening. The reason for the increased surface microhardness is the resulting nanostructured surface layer, which has an average grain size of 40 nm.

### 3.4. Additively Manufactured Materials
### 3.4.1. Inconel 718/Laser Powder Bed Fusion

Kaya et al. [37] studied the effects of the CrA RB process (using the RB method) on the roughness, microhardness, and microstructure of this alloy. A comparison was made with the effects of conventional DR under two other burnishing conditions: drying and preheating at 200 °C. CrA RB achieved the highest integral height 3D roughness parameters. The greatest increase in surface microhardness (compared to the as-built state), or 21%, was provided by CrA RB. Figure 8 illustrates the influence of the burnishing force and burnishing conditions on the obtained surface microhardness, presented in a form normalized to unity. As the force increases, the degree of plastic deformation, and hence the microhardness, increases, whereas the increased amount of heat generated (and imported from outside via preheating) causes the surface microhardness to decrease.

At a depth of approximately 25 μm, CrA RB provides the highest microhardness; at greater depths, the microhardness obtained from CrA RB is the least. The greatest depth of the affected layer is achieved by conventional RB after preheating at 200 °C due to the softening effect. Using X-ray diffraction (XRD) patterns, it was found that CrA RB has the weakest effect on the intensity in the (111) and (200) planes compared to dry RB and RB after preheating the treated surface. After CrA RB, a thin layer of white color was observed on the surface, which is a sign of the nanocrystalline structure of the material.

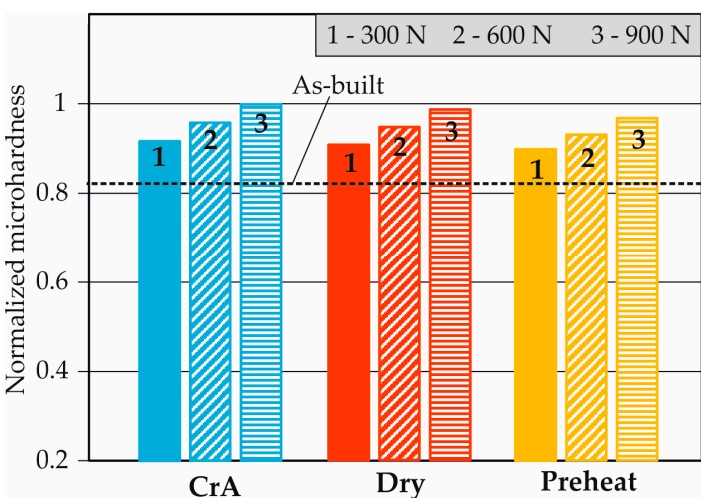

**Figure 8.** Influence of the burnishing force and the burnishing conditions on the obtained surface microhardness of Inconel 718.

### 3.4.2. Laser-Clad Stellite 6 and 420 Stainless Steel as Substrates

Anirudh et al. [59] studied the effects of the CoA low-plasticity burnishing (LPB) process (implemented by the BBHS method) on roughness, microhardness, and residual stresses while maintaining the temperature of the surface at approximately $-40\,^{\circ}$C using liquid nitrogen and a 6 mm ceramic deforming ball coated with silicon nitride. A comparison was made with the effects of conventional LPB and vertical face milling, finding that CoA LPB provides the following: (1) a minimum value of the roughness parameter Sa, with a negative skewness and a kurtosis above three; (2) a maximum surface microhardness; and (3) maximum compressive residual stresses.

### 3.5. Steel

### 3.5.1. Carbon Steel SS 400

CoA (supercritical carbon dioxide + MQL) single-pass SB with a tool rotating around its axis (termed friction stir burnishing) and with tungsten carbide grade K10 insert was performed in several previous studies [38,46,47]. A comparison was made with the effect of SB under dry and MQL conditions, showing that (1) the roughness parameter Ra was reduced by approximately 56.2% and 52.4% relative to dry and MQL conditions, respectively, and (2) the highest microhardness was obtained by cryogenic burnishing close to the surface and up to a depth of 0.27 mm.

### 3.5.2. Tool Steel AISI D3

The CoA DR process (implemented by the BBHS method) has been applied to this steel [39,40] with the aim of SI improvement. Dry ice cooling was applied. A comparison was made against the effects of conventional DR at room temperature. Higher values of the height integral parameters Ra and Rz were obtained relative to conventional DR, which was explained in terms of the degraded roughness resulting from the increased brittleness of the material at lower temperatures, leading to reduced deformability. However, CoA DR was also found to provide a significantly higher hardness (HV0.5) at depths up to approximately 1.4 mm and a significant reduction in residual austenite relative to DR at room temperature.

### 3.5.3. Martensitic Stainless Steel 17-4 PH

CrA DB was implemented by Sachin et al. [50–55] with the objective of improving the SI of this steel. Maximum surface microhardness values of 395 HV, 369 HV, and 357 HV corresponded to maximum surface residual stresses of $-352$ MPa, $-282$ Mpa, and $-195$ MPa, respectively, under cryogenic, MQL, and dry environment conditions [50].

Figure 9 shows the influence of DB conditions on surface axial residual stress and surface microhardness, normalized to unity. Both SI characteristics decrease with increasing heat generation, with the effect being more pronounced for residual stresses.

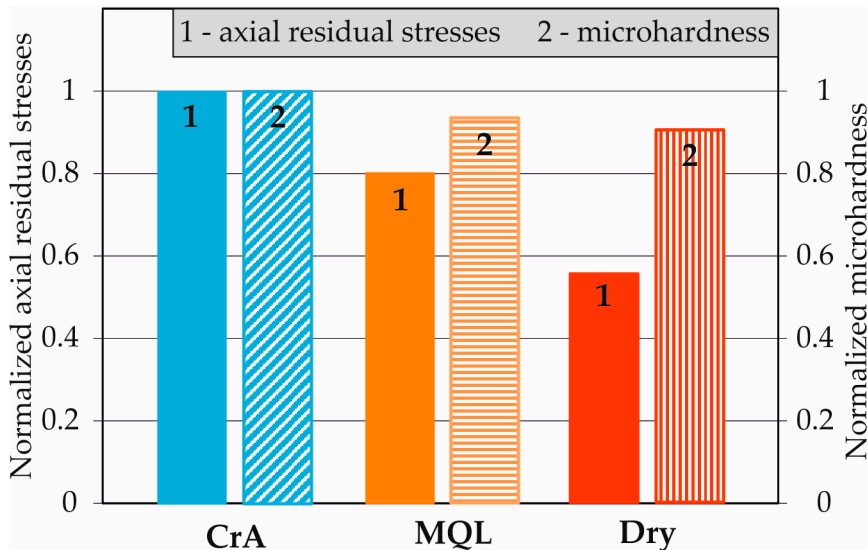

**Figure 9.** Influence of DB conditions on the surface axial residual stress and the surface microhardness of 17-4 PH martensitic stainless steel.

A previous study [51] reported that CrA DB provided a roughness parameter of Ra below 0.14 μm, under which conditions the surface microhardness was increased to 410 HV (340 HV of the base material). In another study [52], it was established that (1) for all investigated combinations of magnitudes of the governing factors, CrA DB exhibited the lowest roughness parameter Ra (below 0.05 μm) and the highest surface microhardness (up to a depth of 120 μm) relative to MQL and dry conditions; (2) the highest compressive surface residual stress of −356 MPa was observed after CrA DB (−298 MPa and −215 MPa under MQL and dry DB conditions); (3) uniform surfaces were formed after CrA DB, whereas MQL and dry conditions resulted in the formation of microdefects. It has also been reported that by means of CrA DB, a roughness Ra = 0.03 μm could be achieved at an optimal burnishing velocity of 85 m/min and a surface microhardness of 413 HV (radius of the deforming diamond insert r = 3 mm) [53]. Using the same radius, an optimal microhardness was achieved at a depth of up to 130 μm, as well as a maximum residual compressive stress of −335 MPa. Similar results were obtained in other studies [54,55].

3.5.4. Austenitic Stainless Steel AISI 304

Nikitin et al. [41] applied the CrA DR process, implemented by the BBHS method, to this austenitic steel, establishing that regardless of the large amount of dissipated generated heat, the cryogenic burnishing condition (−192 °C) leads to a minimum surface residual axial compressive stress of approximately −60 MPa and a positive surface residual hoop stress (close to zero). Increasing the temperature of the processed surface (using halogenic radiant heating) to +300 °C causes the residual axial surface stress to reach −834 MPa (maximum) so that the hoop is approximately −220 MPa. With a further gradual increase in temperature to +550 °C, the absolute value of the axial residual stress decreases to −300 MPa, and the circumferential stress reaches −100 MPa. Nikitin et al. [41] also found that CrA DR produced significantly more (68%) strain-induced $\alpha'$-martensite to depths of approximately 60 μm relative to room temperature conditions (6%). Conventional DR at room temperature forms a nanocrystalline layer with a thickness of 2 μm and grain size of approximately 20 nm. It was reported that with the temperature increasing, grain size increased; however, no information is available to constrain the effect of CrA DR on grain refinement. For the martensitic stainless steel 17-4 PH, a positive effect of CrA DB

on the residual stresses was reported [50–55]. The behavior of the austenitic stainless steel 304 under CrA DR is the inverse [41]. The reason for this difference in behavior is probably rooted in the inherent characteristics of the austenite phase and in the differing contacts between the deforming element and the processed surface, sliding friction [50–55], and rolling friction [41] during the application of the burnishing method.

### 3.6. Biomaterials

#### 3.6.1. Co-Cr-Mo Alloy

The effects of CrA SB on the SI characteristics of Co-Cr-Mo bioalloy specimens were investigated by Yang et al. [60–66]. In one such study [64], the cylindrical specimen was fixed, and one of its flat surfaces was burnished. The deforming roller did not rotate around its axis but rather around the axis of the specimen. The two axes (of the roll and of the specimen) intersected orthogonally. In other studies [60–63,65,66], the high-speed tool steel deforming roller is stationary; the sample rotates around its axis, and the treated surface is cylindrical. In [62], it was shown that CrA SB provided an approximately 40% decrease in the Ra roughness parameter (for two depths of penetration used) relative to dry SB. The authors explain the lower value of Ra with the reduced temperature and the increased wear resistance of the deforming element due to the effective cooling with liquid nitrogen. In other studies [60–63], it was found that CrA SB provided significantly higher microhardness on the surface and at depths of up to approximately 230 μm compared to dry SB. The residual stresses were also investigated in [64], showing that CrA SB with a depth of penetration of 0.127 mm introduces greater axial compressive residual stresses, both at the surface and in depth, than dry SB. As a result, the compressive zone has a greater depth. When the depth of penetration increases to 0.254 mm, CrA SB introduces larger compressive stresses than dry SB at a depth of approximately 50 μm, after which the trend reverses. As a result, the compressive zone introduced by CrA SB is smaller than that of dry SB. The residual hoop stress distribution is the opposite of that described. CrA SB leads to remarkably refined microstructures on the burnished surface [60–63], and grains with sizes below 300 nm have been observed. Compared to dry SB, the depth of the affected layer is increased by up to 170%. An infrared camera was used to measure the temperature and temperature distribution in depth and at the surface, showing that the contact zone temperature was above 700 °C for dry SB but decreased below 300 °C for CrA SB [63]. Lower temperatures inhibit grain growth due to dynamic recrystallization. The latter was studied and modeled in [65]. CrA SB provides two times smaller grains beneath the processed surface than dry SB.

#### 3.6.2. Mg-4Zn-2Sr Alloy

The effects of the CrA RB process (implemented by BB method) on roughness and microhardness were investigated by Akshay et al. [31], who reported a reduction in roughness (roughness parameter not specified) from 3.376 μm to 1.853 μm and an increase in microhardness from 54 HV to 78 HV.

### 3.7. Thermal Spray Coating

Singh et al. [56] applied a CrA SB onto a high-velocity oxy-fuel thermal spray coating (WC-10Co-4Cr) on one face of grit blasted substrate using a rotating carbide-based disc type tool and a workpiece performing translation. The treated surface was planar. These authors have found the following: (1) a reduction in the average roughness of the coating from 1.94 μm to 0.84 μm; (2) an increased surface microhardness; (3) an isotropic region of surface residual compressive stresses greater than those introduced by dry SB; and (4) reduced porosity within the surface layer.

## 4. Effects of CrA and CoA Burnishing on Operating Behavior

The effects of the hybrid processes on the operating behavior are analyzed for each of the studied groups of materials.

### 4.1. Magnesium Alloy AZ31B-O

The effect of CrA SB on the corrosion resistance of this magnesium alloy was studied by Pu et al. [42–45], who found a significant improvement in corrosion resistance due to smaller grain size. Near the uppermost surface, the grain size reduces to 523 nm, and the smallest size reaches up to 263 nm. The reason for this small size is the strain-induced dynamic recrystallization and effective cooling with liquid nitrogen, which prevents grain growth.

### 4.2. Titanium Alloy Ti-6Al-4V

Rotella et al. [48,49] studied the effects of the CrA DR process on the wear resistance of this alloy via a linearly reciprocating ball-on-flat sliding wear test. A comparison was made to determine the effects of DR under dry and MQL conditions, showing that the lowest wear rate of the burnished surface occurred after CrA DR. The favorable combination of cryogenic conditions and the use of a coated deforming tool lead to the best surface hardness (an increase of about 60% compared to dry and MQL conditions) and, thus, the highest wear resistance.

The effect of CrA SB on the corrosion resistance of this alloy was investigated by Tang et al. [57,58], who demonstrated its increased corrosion resistance due to the formation of a more rapid, stable, and less defective passive film over the surface nanocrystalline layer due to its high density of grain boundaries and dislocations.

### 4.3. Austenitic Stainless Steel AISI 304

Nikitin et al. [41] studied the effects of the CrA DR process (using the BBHS method) on the fatigue behavior of specimens made of this steel and made a comparison with the effects of DR at temperatures between −192 °C and +800 °C. The rotary bending fatigue test was conducted at room temperature (25 °C), and the cycle parameters include the cycle asymmetry factor R = −1 and a stress amplitude of 360 MPa. These authors have established that CrA DR led to the lowest number of cycles before fatigue failure. This was interpreted to be due to the large content of strain-induced $\alpha'$-martensite in the surface and near-subsurface layers. The longest fatigue life (48,000 cycles) was shown by the samples heated under burnishing at 550 °C. The influence of the temperature during the DR process on the surface axial residual stresses and the corresponding number of cycles to failure in a normalized form to unity is shown in Figure 10, which clearly shows that the maximum surface compressive residual stress does not automatically lead to a maximum number of cycles to failure.

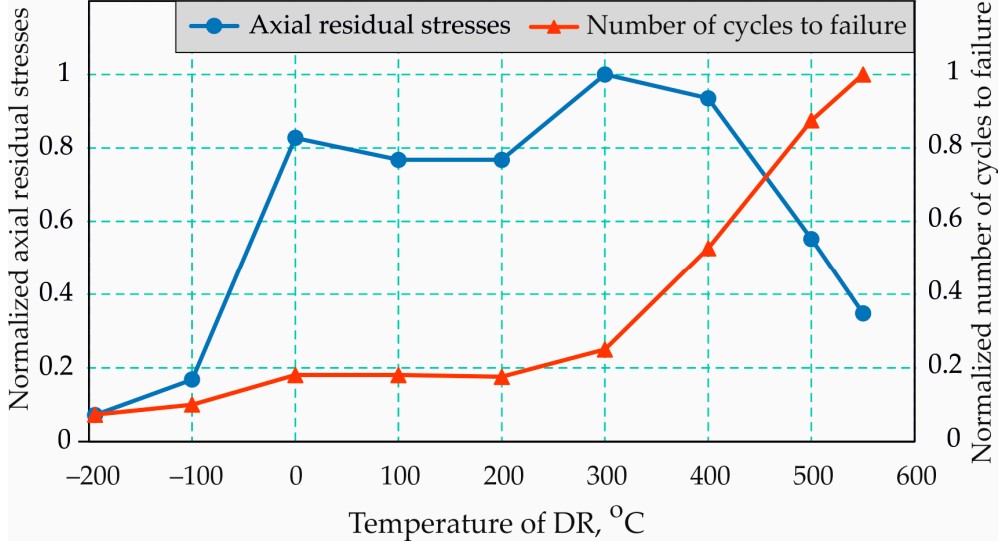

**Figure 10.** Influence of the temperature during the DR process on the surface axial residual stresses and the corresponding number of cycles to failure of 304 austenitic stainless steel.

The authors' explanation [41] is that at a temperature of 550 °C, the strain aging effect occurs, which multiplies the conventional coldworking effect. As a result, the most favorable combination of SI characteristics is obtained: an absence of martensite; formation of carbides; near-surface nanocrystallisation; high dislocation densities; surface quality; and (to a lesser extent) compressive residual stresses. The S-N curves constructed by these authors (Figure 11) show that the phenomenon described above is valid over the entire high-cycle fatigue field.

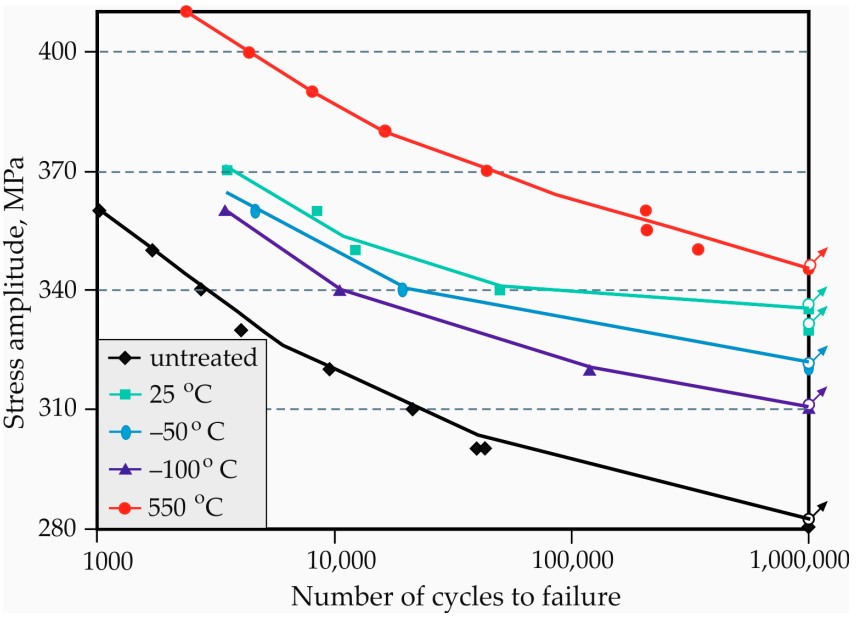

**Figure 11.** S-N curves of 304 austenitic stainless steel [41].

*4.4. Biomaterial Co-Cr-Mo*

Yang et al. [64] subjected this alloy to CrA SB to increase its wear resistance using a pin-on-disk tester, finding that the cryogenic condition provides minimal wear volume losses compared to SB under dry conditions. The authors explain this phenomenon by microstructure refinement, compressive residual stresses, and the occurrence of a preferred hexagonal close-packed phase; moreover, this phase has been revealed to be the property with the greatest degree of influence over wear resistance.

**5. Conclusions and Future Research Perspectives**

This review paper has summarized the state-of-the-art hybrid processes of CrA and CoA burnishing, allowing for the following conclusions to be drawn:

- For all investigated materials, the CrA and CoA burnishing processes significantly increase surface microhardness relative to other burnishing conditions. This is a direct consequence of the nanostructured surface layer produced. The dynamic recrystallization induced by surface plastic deformation takes place under intense cooling conditions, which inhibit grain growth;

- The benefit from the hybrid processes to improve the roughness compared to the effect of burnishing under other conditions (dry, flood, MQL, preheating), as well as compared to other finishings (for instance, grinding), is not unambiguously defined for the different materials and burnishing methods. For the magnesium alloy AZ31 B-O, grinding at room temperature achieves a lower value of the Ra roughness parameter than CrA SB. For the Ti-6Al-4V alloy, the CrA DR process realized by the RB method achieves a higher value of the Ra parameter than DR under other conditions, but the hybrid CrA SB process reduces Ra relative to dry SB. The roughness worsens when CrA DR is applied to Inconel 718 and tool steel AISI D3. In the remaining analyzed cases (Table 2), hybrid CrA and CoA processes improve the resulting roughness. It

is important to note that when the hybrid process is implemented using SB or DB methods, the roughness is always improved relative to SB (respectively DB) under dry, flood, and MQL conditions;

- With the exception of CrA DR using 304 austenitic stainless steel [41], hybrid processes increase the compressive residual stresses in the surface layer. Intensive cooling greatly reduces the thermal effect, leading to softening. As a result, the surface microhardness and surface residual compressive stresses increase. The resulting effect is analogous to the effect of burnishing hardened steel;
- CrA burnishing increases wear and corrosion resistance because the nanostructured surface layer exhibits increased microhardness;
- Only one study has considered the effects of CrA and CoA burnishing on fatigue behavior. CrA DR (implemented via the BBHS method) was applied to austenitic stainless steel 304 [41], showing that CrA DR leads to the lowest number of cycles to fatigue failure relative to cases in which DR is conducted on preheated surfaces. This occurs as a result of the large content of strain-induced $\alpha'$-martensite in the surface and near-subsurface layers.

Based on the analyses conducted and the conclusions outlined above, the following directions for future investigations can be proposed:

- It is of interest to study the effects of CrA and CoA burnishing on the SI and operating behavior of important material groups, such as aluminum bronzes, which are widely used in the marine industry, shipbuilding, aviation, railway, offshore platform applications, and other fields. For example, single-phase aluminum bronzes do not undergo heat treatment, and their surface microhardness may only be increased by surface cold working. It is, therefore, of interest for engineering practices to determine the effects of burnishing under cryogenic or cool conditions with the objective of improving the microstructure and surface microhardness of bronzes, in addition to improving bronzes with β-transformation;
- The effects of CrA and CoA burnishing on SI and the operating behavior of high-strength aluminum alloys, which are widely used in the aerospace and automotive industries, have not been fully evaluated. The only exception is 7050-T7451 AA [35,36]; but for this alloy, no information has been reported regarding the effect of CrA and CoA burnishing on its operational behavior;
- Chromium–nickel austenitic stainless steels form 70% of the total share of stainless steels used for various industrial applications. However, only one study has considered the CrA DR of AISI 304 steel [41]. Other studies have shown a significant increase in the mechanical characteristics of such steels when subjected to cryogenic temperatures [8,9]. Therefore, the effect of conventional surface cold working on austenitic steels that have been previously subjected to autonomous cryogenic treatment remains of interest;
- There is a lack of research on the effect of the cryogenic conditions during the burnishing process on the dimensional accuracy of the treated surfaces. The results of such research are essential for engineering practice since burnishing is a finishing process;
- Increasing the fatigue life of metal components is essential to engineering practice. Therefore, investigating the effect of CrA and CoA burnishing on the fatigue behavior of metallic components would be of great interest.

**Author Contributions:** Conceptualization, J.M. and G.D.; methodology, J.M. and G.D.; software, J.M. and G.D.; validation, J.M. and G.D.; formal analysis, G.D. and J.M.; investigation, G.D. and J.M.; resources, J.M. and G.D.; data curation, J.M. and G.D.; writing—original draft preparation, J.M. and G.D.; writing—review and editing, J.M. and G.D.; visualization, J.M. and G.D.; supervision, J.M.; project administration, J.M. and G.D.; funding acquisition, J.M. and G.D. All authors have read and agreed to the published version of this manuscript.

**Funding:** This research was supported by the European Regional Development Fund within the OP. "Science and Education for Smart Growth 2014–2020", Project CoC "Smart Mechatronics, Eco- and Energy Saving Systems and Technologies", No. BG05M2OP001-1.002-0023.

**Data Availability Statement:** The original contributions presented in this study are included in the article; further inquiries can be directed to the corresponding authors.

**Conflicts of Interest:** The authors declare no conflicts of interest.

## Abbreviations

| | |
|---|---|
| BB | ball burnishing |
| BBHS | ball burnishing with hydrostatic sphere |
| CrT | cryogenic treatment |
| CoA | cool-assisted |
| CoAB | cool-assisted burnishing |
| CR | corrosion resistance |
| CrA | cryogenic-assisted |
| CrAB | cryogenic-assisted burnishing |
| DB | diamond burnishing |
| DR | deep rolling |
| F | fatigue |
| LPB | low plasticity burnishing |
| M | microstructure |
| MH | microhardness |
| MQL | minimum quantity of lubrication |
| R | roughness |
| RB | roller burnishing |
| RS | residual stresses |
| SB | slide burnishing |
| SI | surface integrity |
| WR | wear resistance |
| XRD | X-ray diffraction |

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
