# Peer review of "Effects of Cryogenic- and Cool-Assisted Burnishing on the Surface Integrity and Operating Behavior of Metal Components: A Review and Perspectives"

_machines, doi:10.3390/machines12050312_

Round 1
Reviewer 1 Report
Comments and Suggestions for Authors
The is a review article focusing on cryogenic- or cool-assisted burnishing of metals. The number of references is not enough for a review paper with such a wide topic with long history. Massive citations of references from the same author and similar research can be found, which is not appropriate. For instance, [43-45], [50-55], [60-66]. The authors need to include more studies in this field from different groups and countries, otherwise, Figs 4-6 are meaningless. Also, the references are relatively old, only 49 % (42/86) of them are post-2016, 15 % (13/86) are post-2020. Another issue is that the authors simply discussed the experiment results from literature, without insights and discussion of any models, theories or mechanisms. It reads more like a survey report.
My other comments are list as follows.
1. The abstract only mentioned the importance of cryogenic treatment while another key issue, burnishing, is missing. This manufacturing process and its importance should be emphasized in the abstract.
2. The list of abbreviations should be placed at the first page before Introduction to make it more readable.
3. In Fig. 3, the pie chart, i.e. the colour background is not necessary. It is relatively misleading for the readers that the pie chart represents the portion of these methods in industries.
4. The meaning of LN2 is not given in this paper.
5. In Tables 1 and 2, it is noted that √* is an indicator of improvement, what is the meaning of √?
6. In Figs. 7-10, why the authors normalise the indicators? The data are just from the same material and the same research. And the authors did not give the criteria of normalisation.
7. The Conclusion part needs to be rewritten. The authors are still discussing the results case by case rather than pointing out any conclusions.
Comments on the Quality of English Language
Can be improved.
Reviewer 2 Report
Comments and Suggestions for Authors
The presented paper deals with the review of cryogenic- and cool-assisted burnishing of different materials. The authors analyse 86 articles, 11 of which are review type.
The review article is written at a high scientific level. However, there are some small issues which could be improved:
- in the whole manuscript, you are using CrT as an abbreviation for cryogenic treatment. However, you used the abbreviation CT in Fig. 2 for the same meaning.
- in Tab. 2, you used the abbreviation M (for microstructure), but the M is not explained under the table.
- in line 509 where you mentioned martensite type, there is just some kind of spiral instead of a proper symbol.
- it could be interesting to analyse cryogenic burnishing in terms of final accuracy due to the material´s thermal expansion since it is used as a finishing method.
Reviewer 3 Report
Comments and Suggestions for Authors
A brief summary:
The aim of the paper was to conduct a review on cryogenic- and cool-assisted burnishing. Authors analyzed the effect of adopting low-temperature to those processes on the surface integrity and operating behavior of various metallic materials. Authors investigated different cryogenic- and cool-assisted burnishing processes and how its use affect various alloys. The review provides insight on the current state of research on CrAB and CoAB as well as indicates the possibilities of future research developments.
Broad comments:
A significant achievement of the presented work is conducting a comprehensive review on different cryogenic- and cool-assisted burnishing. The authors presented general overview in form of two comprehensible tables as well as detailed description illustrated with graphs with normalized parameter values. Based on the review authors concluded the main reasons to adopt CrAB and CoAB. The literature consists of 86 papers, significant amount of which were published in the last few years. The editing of the paper is mostly fine, although there are some typos that should be addressed.
Specific comments:
· Line 29: authors refer to “negative temperatures” without specifying the unit, which indicates temperatures below absolute zero. Authors should add a unit or rephrase.
· Line 417: authors refer to a thermal spray coating. What is the material of the coating? Is it Mg-4Zn-2Sr alloy and if so, what is the base material?
· Paragraph 4.1 and 4.2: authors use many vague phrases, such as “significant improvement”, “the best surface hardness”, “increased corrosion resistance”. Authors should be more precise and use some specific data.
· In Conclusions authors specify the directions for future investigations. In the previous paragraph authors mentioned that there is only one study considering the effects of CrA and CoA burnishing on fatigue behavior. Authors could include the topic as a possible future investigation direction.
· There are some minor editing errors and typos that should be addressed, i.e. line 119 “…burnishing. using the ball…”, line 334 “austeniterelative”, line 193 “…Grain refinement occurs be mechanism…” or Figures 7, 8, 9 “microhrdness”,
Comments on the Quality of English Language
The language of the paper is mostly fine, however there are some errors that should be corrected, i.e. line 26 “…treatments….consists”, line 202 “…The effects…was studied…”etc. In addition, authors tend to mix tenses i.e. line 156 “…process is…”, line 157 “…process was…”
Round 2
Reviewer 1 Report
Comments and Suggestions for Authors
No